# The Concepts of Women’s Empowerment in Child Malnutrition Programs in Luangprabang Province, Lao People’s Democratic Republic

**DOI:** 10.3390/ijerph20176662

**Published:** 2023-08-28

**Authors:** Kanchana Thilakoun, Daniel Reinharz, Sengchanh Kounnavong

**Affiliations:** 1Department of Social and Preventive Medicine, Laval University, Laval, QC G1V 0A6, Canada; kanchana.thilakoun.1@ulaval.ca; 2Lao Tropical and Public Health Institute, Vientiane 01030, Laos; skounnavong@gmail.com

**Keywords:** women’s empowerment, child malnutrition, internal organizations, external organizations, Lao PDR

## Abstract

In several developing countries, such as Lao People’s Democratic Republic (Lao PDR), the fight against malnutrition is carried out through programs that involve collaboration between internal (national) and external (international) actors. These actors may have different perceptions on what is one of the pillars of these programs: the empowerment of women, especially mothers of young children. Little is known about these differences and the impact of these differences on the empowerment component of collaborative projects and the perception of its impact on the reduction in malnutrition in the country. A multiple case study was performed. Data collection was carried out in Vientiane Capital and Luangprabang province. The data were obtained from (1) documents, (2) semi-structured interviews with representatives of internal and external organizations, and (3) focus group discussions and individual interviews with mothers of children under five years old. Analysis consisted of characterizing the empowerment component of nutrition programs of internal and external organizations, as well as mothers, based on an OXFAM’s adapted conceptual framework on women’s economic empowerment. The study revealed a common understanding among government and external organizations regarding the significance of promoting women’s empowerment for reducing child malnutrition in Lao PDR. However, variations were observed in the interpretation of specific determinants of women’s empowerment, specifically in relation to women’s autonomy and the role of social capital. The perspective of internal actors includes the political ideology and traditions that make Lao PDR a distinct country. This perspective dominates the nutrition programs conducted under the collaboration of internal and external actors. In Lao PDR, the concept of women’s empowerment in nutrition programs conducted through collaboration between internal and external actors and targeting young Lao mothers gives prominence to political and socio-cultural factors.

## 1. Introduction

Child malnutrition in Lao PDR, although improving, it is still high compared to neighboring countries, such as Thailand and Vietnam. The latest data on child malnutrition show that the stunting rate among children under five years of age was 31.5% in 2021 [1]. This number is 1.6 times higher than in Vietnam (19.6% in 2020) and 2.3 times higher than in Thailand (13.4% in 2019) [2]. Nearly half of all deaths in this age group are linked to malnutrition [3]. Malnutrition is not only a determinant of child mortality and morbidity, it is also a cause of poverty by its deleterious effect on physical and cognitive development. It therefore perpetuates missed socio-economic opportunities throughout the course of life and across generations [4]. Malnutrition in children under 5 years old is, therefore, a public health priority [4].

The backbone of the fight against malnutrition in Lao PDR is the National Nutrition Policy (NNP) enacted in December 2008 [5] and a series of National Plans of Action on Nutrition (NPAN). The NNP and NPAN aim to support actions that target the full range of malnutrition determinants [6]. Among these determinants is the empowerment of women, particularly mothers of young children [7,8,9,10,11]. Indeed, the NPAN identify empowering women as a key strategic intervention objective. For example, strategy SO13 of the NPAN 2021–2025 states explicitly the need to work on “promoting gender equality” and paying specific attention to ensuring the balance between women’s empowerment, women’s involvement in production, and women’s role in child raising [4]. The importance given to women’s empowerment is consistent with the scientific literature. Many studies have shown that women’s empowerment plays a key role in addressing child malnutrition. Recognizing the role of women’s empowerment is pivotal to effectively fight against child malnutrition [12,13,14,15,16,17,18,19,20,21]. 

The concept of empowerment in public health programs, such as those targeting malnutrition, has been little studied in single-party developing countries like Lao PDR. Indeed, Lao PDR has a single-party political system. The concept of women’s empowerment that prevails in this country is therefore rooted in the sense that the party has a responsibility to liberate traditionally oppressed groups, including women, from their oppressed state [22]. This rationale explains the predominance of top-down approaches found in state interventions that aim to increase communities’ empowerment [22,23]. In contrast, in countries that have a multiparty political system, the concept of women’s empowerment is embedded in a history of social demands by marginal communities [24,25]. This rationale explains the importance of bottom-up approaches found in interventions that aim to increase the empowerment of communities, including women [26].

When organizations promoting these two archetypes of the empowerment concept must work together, they have to compromise on which aspects of the concept to support and which to develop. This is the case in Lao PDR, where most interventions against child malnutrition involve collaboration between government and external organizations who are expected to interpret the concept of women’s empowerment differently. How these expected differences influence activities promoting women’s empowerment, which are found in most anti-malnutrition programs, has never been explored. Therefore, nothing is known about how these differences need to be taken into account to build activities on the ground that are culturally acceptable and effective in reducing malnutrition in the country.

This article aims to characterize the convergences and divergences of the concept of empowerment between internal and external organizations collaborating in projects to combat malnutrition in Lao PDR, and to describe the effect on the empowerment component of projects targeting mothers and children under the age of five.

## 2. Materials and Methods

A multiple case study was performed. Information came from interviews and focus groups, as well as from relevant documents on nutrition and empowerment.

### 2.1. Conceptual Frameworks

Figure 1 shows the conceptual framework on which the study is based. The three major groups considered are (1) government and party organizations, (2) non-governmental organizations (NGOs), and (3) other organizations as UN Agencies, which interact to implement programs to combat malnutrition. Each has its own conception of empowerment. Their interactions lead to a peculiar concept of empowerment that will be transmitted to the target populations of the interventions, i.e., mothers of children under 5 years old in the community. 

The concept of empowerment specific to stakeholders is described through the dimensions of the Oxfam’s Conceptual Framework on Women’s Economic Empowerment [27]. This framework has been used in many studies carried out in Southeast Asia (SEA) in the field of nutrition and food security [28,29,30], although never to study the emergence of a concept that emerges from the confrontation of several different conceptions. Yet, the framework has been applied in numerous studies across SEA, especially to measure women’s empowerment. However, as suggested by the inventors of the framework [27], items of the conceptual framework that were considered as not directly relevant to our research topic or too sensitive to be discussed, such as opinions on property rights, political participation, and control over sexuality, were removed. We also changed the “power in markets” item into “power in economic pursuits”, as we believe the latter covers a broader aspect of improving women’s income. Items considered are categorized into three main groups: the personal, relational, and environmental domains. The personal domain refers to changes at the individual level, notably regarding self-esteem, knowledge, opinions on gender rights, personal autonomy regarding the decision to act independently, and autonomy around violence against women. The relational dimension focuses on changes at the group level, covering group participation, household decision making, control over assets, and social capital. The environmental dimension refers to changes at the community level, encompassing accessibility to legal and healthcare services, safety of movement, social norms, ability to influence at the political level, and advocating for change for women.

### 2.2. Study Sites

Data collection was conducted in two provinces: Vientiane Capital and Luangprabang. Vientiane Capital is where the headquarters of ministries and other government organizations, as well as those of NGOs, are located. Luangprabang province has a longstanding history of implementing nutrition programs that require collaboration between internal and external stakeholders. Presently, there are over 20 programs operating throughout the province, with at least 10 programs specifically implemented in Luangprabang district. These initiatives involve both local and external organizations working together to address the issue of child malnutrition [31]. The selection of villages in the province of Luangprabang was made in collaboration with the provincial Lao Women Union (LWU). The LWU provided two lists of villages where nutrition programs are being implemented in Luangprabang province. The first list comprises villages where the Women’s Workload Reduction/Gender Equality in a Relationship (WWR/GER) and VSLA initiatives are being carried out. The second list consists of villages where the 1000-days program is being implemented. From each list, we randomly selected one village. 

### 2.3. Study Population

The population consisted of (1) representatives of organizations in Vientiane Capital and Luangprabang province that are involved in the fight against child malnutrition. These participants have been working in their respective organizations for at least one year; (2) Mothers of under-five-year-old children from two different villages that have benefited from two different nutrition collaborative projects, namely the GER/WWR in Sen Souk village and the 1000-day household visits project in Vangneun village. These projects involved both internal organizations, such as government and party organizations, and external organizations, such as international NGOs [1,4].

### 2.4. Data Collection Procedures

A preliminary list of organizations involved in child malnutrition research projects in Lao PDR was drawn up with the assistance of the Lao Tropical and Public Health Institute (Lao TPHI), the main organization conducting such research studies in the country. Directors of these organizations received an invitation letter seeking permission to interview a representative. Upon permission, they identified suitable participants. The suggested individuals were contacted via phone. They were asked to take part in a face-to-face interview or by Zoom or WhatsApp, at their convenience. Those who agreed were then emailed a consent form to be signed and returned before the meeting took place, at a time and place deemed appropriate by the respondent. At the end of the interview, participants were asked to propose names of people who might bring complementary or different point of views about the research topic. The researcher then directly contacted the director of the organizations of the suggested names via email and WhatsApp.

Mothers of children aged under five years old in villages were recruited with the support of the village’s LWU heads. In village one, recruitment was conducted during a VSLA monthly meeting held at the village office, attended by representatives of 72 families participating in the VSLA program. Seven mothers with under-five-year-old children were randomly proposed to participate in the study and willingly agreed after the researcher explained the purpose and procedures of focus group discussions (FGDs). The FGD took place on the same day, following the VSLA meeting, with only these seven mothers staying behind for the discussions. In village two, the village’s LWU head contacted mothers of children under five years old via phone a day before the FGD and invited them to participate. The next day, the researcher met with the LWU head at the village meeting place, where eight mothers with under-five-year-old children were present and invited to take part. The researcher explained the study’s purpose and FGD procedures, and all eight mothers expressed their willingness to participate. Later in the afternoon, they gathered at the same place for the FGD. 

After the FGDs, three mothers were randomly selected for individual interviews conducted by phone about two weeks later, and all of them agreed to be interviewed. Participants signed a consent form before the individual interviews and the FGDs started. 

The duration of individual interviews and the FGDs was about 60–90 min. Data collection was conducted between August 2022 and January 2023. 

### 2.5. Data Collection Tool

Data collections tools consisted of (1) an interview guide for representatives of organizations and (2) a discussion guide to be used with mothers either in focus groups or in individual interviews. Both tools were constructed with the participation of a member of the Lao Tropical and Public Health Institute (SK) who has extensive experience in conducting nutrition research projects in the country, so that each dimension of the framework reflected the socio-cultural particularity of the study context. This is in accordance with the recommendations of the inventors of the conceptual framework [27].

The themes of the interview guide aimed to retrieve information on the dimensions of the conceptual frameworks of the study. The main themes were related to (1) the stakeholders’ concepts of women’s empowerment; (2) the translation of these definitions into the actions implemented in the programs run by the stakeholders in the fight against malnutrition; (3) and the perception of the existence of different perceptions of the concept of women’s empowerment by collaborators in a program and its impact on the capacity of the program to improve child-feeding practices and the fight against child malnutrition. 

The individual interviews and FGDs with mothers at the community level were based on the second tool, i.e., the discussion guide. This guide aimed to retrieve (1) the mother’s knowledge and perception regarding child-feeding practices before and after the implementation of nutrition programs and (2) the mother’s perception regarding their ability to feed their children.

The interview guide and the discussion guide were pre-tested to ensure their comprehensibility before data collection with individuals representing the target groups (3 professionals working in the field of nutrition and 3 villagers), who were not part of the sample.

In order to gain more information, the documents related to women’s empowerment and the fight against child malnutrition in Lao PDR were also searched in the following databases: Library of Laval University; PubMed; ScienceDirect; and references listed in published articles, websites, and Facebook pages of the government ministries, the party, UN agency, civil society, EU, and NGOs in Lao PDR. Some documents were directly provided by participants.

### 2.6. Data Management

Interviews and FGDs were transcribed on the same day in the language of the interview. Interviews in Lao were also translated into English. Then, the transcriptions and notes about the meeting were sent the same day to one of the coresearchers for him to make comments before the next interview or meeting took place. The comments were intended to provide suggestions on aspects to eventually explore more deeply in the next meeting.

### 2.7. Data Analysis 

A deductive–inductive approach was used in our analysis, which was carried out using NVivo 12. Initial codes were generated from the data and then organized into themes and categories based on our conceptual framework. The themes and categories were then compared across cases to identify similarities and differences in the characteristics of women’s empowerment concepts.

The validity of the analysis was based on four criteria proposed by Lincoln and Guba [32]: (1) credibility (triangulation of information); (2) transferability (detailed description of the context); (3) dependability (analyses carried out independently by the researcher and one of her supervisors, and search for a consensus between them); and (4) confirmability (translations, results, interpretations, and conclusions supervised by a Lao supervisor).

Interviews and documents were coded independently by the researcher and a coresearcher. Consensus meetings were held to discuss coding inconsistencies, with the second coresearcher consulted in case of disagreement. A few audits with some participants were conducted by phone, asking them to judge the accuracy of their words in the transcriptions. Changes were made based on their feedback. An audit and verification by the second coresearcher were also performed to ensure the confirmability.

## 3. Results

A total of 32 participants were interviewed. Of our 32 participants, 10 were the representatives of the internal organizations (government or party organizations), 7 were representatives of the external organizations, and 15 were mothers of under-five-year-old children from two villages. In addition, six documents were analyzed: (1) the NPAN 2021–2025; (2) the National Nutrition Policy; (3) the LWU’s law on the protection and promotion of Lao women’s rights; (4) the official document on the “Three Good Deeds” of the Lao women’s Union; and (5) the End Evaluation Final Report on Partnership for Improved Nutrition in Lao PDR Pillar 3: Sustainable Change Achieved through Linking Improved Nutrition and Governance (SCALING) (Table 1).

### 3.1. Concept of Women’s Empowerment in Lao PDR

Table 2 shows the presence of the different items of the concept of women’s empowerment among the three different stakeholder groups: internal organizations, external organizations, and mothers of under-five-year-old children. The table categorizes the items in the three dimension of the women’s empowerment concept: individual, relational, and environmental dimension. 

#### 3.1.1. Individual Dimension

The individual dimension of the women’s empowerment concept relates to changes at the personal level. Table 2 shows that among the twelve items, nine are presented in all three stakeholders’ empowerment concept (“self-esteem, self-confidence, and self-efficacy”; “individual knowledge”; “opinions on gender rights”; “opinions on power within the household”; “opinions on freedom of movement”; “recognition of care”; “individual capability”; “access to savings”; and “access to credit”). 

This finding highlights the importance of knowledge and improving leadership skills as pillars of the concept of empowerment. Developing these components requires respondents and an array of strategies. For example, health education is increased through the use of different activities, notably antenatal care (ANC) in healthcare centers, 1000-day household visits, community education, and mass media. The information provided to mothers or women is mostly related to self-care during pregnancy, vaccination, family planning, breastfeeding, complementary food, child nutrition, and mother’s nutrition. Mothers confirmed the importance of knowledge. The acquisition of knowledge about child nutrition is felt as a key factor that increased their capacity to take care of children. Regarding leadership skills, all stakeholders highlight that improving this characteristic promotes the presence of the nine common items.

However, it is noted that the presence of elements does not necessarily mean the same understanding of their definition and purpose as those of other groups. Government officials, particularly the LWU, see their function in relation to women’s empowerment as having to build on these items to promote the concept of “Three Good Deeds” (ຂໍ້ແຂ່ງຂັນແມ່ຍິງ 3 ດີ). The “Three Good Deeds” concept encompasses three main aspects: good citizens; good development; and building good and prosperous families. Firstly, being a good citizen entails patriotism and adherence to the political one-party system, as well as having a family and social relations in accordance with Lao culture as promoted by the party. Secondly, being a well-developed woman involves displaying party-defined revolutionary moral values and committing oneself to contribute to generating income for oneself and the family. Lastly, building a harmonious family environment means working to create a model family that respects the party’s vision of Lao society. 

Of the twelve items presented in Table 2, two items are found in the empowerment concept of internal and external organizations (“non-acceptability of gender-based violence (GBV)” and “personal autonomy regarding violence against women (VAW)”). Representatives of both internal and external organizations emphasize the importance of addressing gender-based violence and protecting women from socio-cultural and traditional practices that oppress women, such as child marriage and girls’ school dropouts. Addressing this issue therefore contributes to long-term empowerment. None of the mothers with children under the age of five mentioned these specific items.

In addition, internal and external actors also highlight the importance of promoting personal autonomy to act independently for women. However, there is a notable divergence in how the two groups understand and apply this particular item. The following quotes illustrate the divergence in understanding of “autonomy to act independently.” Internal actors believe in supporting women by providing guidance and leadership, whereas external actors view it as empowering women to assert themselves, raise their voices, and challenge certain societal norms and practices:

Internal actors:

“*In nutrition work, we have the leadership activity. We teach them even how to talk, how to communicate, how to be a leader. So, we teach them all.*”Participant 4.

“*[…] to support women to be independent […] We have to train them, we have to train the family, both husband and wife […] we have to provide them with knowledge. The more we provide them with knowledge, the better it is.*”Participant 12.

External actors:

“… *I can say that women have to take action on their own lives and then she can support others to take that action in their lives […] So not only building their capacity, but also making them responsible, accountable, giving them position, role and taking them in the leadership position*”Participant 13.

“*[…] younger female team members not only understand that they “can” give their opinion, but they “should”. […] Because sometimes “they can” but they don’t think “they should” and that’s very common because there’s that “appropriateness […], we can change education, we can have knowledge, but do we, do we “ກ້າ (dare)” to speak up and do we think it’s appropriate to speak out?*”Participant 8.

#### 3.1.2. Relational Dimension

The relational dimension of the women’s empowerment concept relates to changes at the relational level, which refers to changes taking place in power dynamics between a woman and other household members. Table 2 shows that out of sixteen items, nine are found in every stakeholders’ empowerment concept (“group participation”; “level of support provided by groups to pursue own initiatives”; “involvement in household decision making (expenditure, investment, and household management decisions)”; “control over household assets”; “contribution to household income: independent income”; “control over time and workload”; “ability to reduce time devoted to care responsibilities”; “ability to redistribute the burden of care responsibilities”; and “ability to have more time for leisure and to socialize”).

To enhance these nine aspects, the respondents mentioned the importance of women’s communication skills, support from close individuals, particularly husbands, and access by women to savings and credit. The internal and external actors highlight that a woman is empowered if she can communicate within the family. Communication is a determinant of involvement in household decision making and in the request for help regarding household work chores. To facilitate this aspect, the government and the external actors work closely on GER/WWR interventions. These interventions that highlight communication abilities, target not only women but also their husbands. According to the FGD, mothers recognize that GER/WWR activities have a positive impact on their participation in household decision making. The mothers also mentioned that these activities help them to improve their family relationships and to gain support from the people around them, including their husbands and their children. These allow them to have more time to take care of themselves and their under-five-year-old children. They also have more time to meet their friends and exchange their experiences and knowledge. Furthermore, the results suggest a connection between these findings and two additional items that were exclusively mentioned by mothers: participation in public events and power in economic pursuits (such as power in markets and local business). Support from husbands enables women to participate in public events and dedicate more time to their household business endeavors.

Representatives of the government staff and the external actors also point out that increasing women’s control over household savings and credits through the VSLA initiative is an important key to empowering women. According to our three groups of actors, this intervention allows women to not only easily access to their savings, but also to increase their ability to involve household assets in decision making (including the ability to control household assets) and be able to contribute to household income. Moreover, the VSLA also influences the husbands’ attitudes toward their wives. The Scaling Endline Evaluation Report showed that participation in a VSLA makes women more independent in financial decision making and more respected by their husbands. Throughout the researcher’s observation during the VSLA activity, it was noticed that the activity can also increase the women’s ability to make decisions at the community level. Women participating in the VSLA have to meet every month to deposit their money and make decisions all together on who could borrow the money from the association. 

In order to effectively implement interventions aimed at women’s empowerment, it has been observed that the support of village authorities is crucial. These include the support of the heads of the villages, village LWU, and ethnic minority leaders. Internal and external actors stress that they have to engage village authorities in the implementation of their interventions. Women feel secure when messages are conveyed by people they know and trust, notably the village heads or village LWU members. These individuals are viewed as role models by the mothers. 

Respondents from the government and external organizations that are intersectoral collaborations are the main determinants of these nine items. These collaborations involve actors from health, agriculture, education, local LWU, and authorities at provincial, district, and village levels. According to the respondents, interventions aiming to empower individuals should not only target the beneficiaries of health and social programs, but also the personnel of organizations concerned, whatever sector they refer to. 

Table 2 shows that four items not mentioned by women are present in the concepts of internal and external actors (“attitudes and beliefs of people close to the woman”, “degrees of influencing in community groups”; “contribution to community social needs”; and “experience of gender-based violence (GBV). In order to be able to influence the attitudes and belief of the very close person to the women (husbands, family, and the community), both groups stress that it is important to provide knowledge related to child nutrition and gender equality to these people, especially through 1000-day household visits, community health education, GER/WWR activities, and mass media. When the husbands, family, and community have a better understanding of the importance of child nutrition and see their values in supporting women, women have a supportive environment to implement healthier child-feeding practices. Representatives of external organizations highlight that improving the knowledge of people close to women is considered important because they are typically influential and respected in the household and village. If these individuals can be convinced to adopt new attitudes and behaviors, they can become key actors of behavioral change and help to shift social norms to support empowering women in their community.

In addition, representatives of internal and external organizations agree that women are empowered if they are able to influence the community and contribute to community social needs. The external actors also stress that when women are empowered, they can advocate for their own needs and the needs of other women and be role models and inspiration for other women.

Internal and external actors also recognize GBV as an important component of their concepts. They concur that addressing GBV and VAW is crucial for empowering women. A common understanding among these actors is that achieving gender equality is fundamental, encompassing addressing power dynamics between genders at various levels, including households, workplaces, and society. Furthermore, there is a shared agreement that gender mainstreaming should be integrated into organizational policies and practices, including those of government and external organizations. Both actors agree that setting targets to increase the representation of women in organizations and implementing gender equality policies are crucial steps towards enhancing women’s empowerment and representation. However, both actors hold differing perspectives on gender equality, reflecting variations in their understanding. While both find common ground in advocating for increased women’s representation, the external actors emphasize that discrimination against women persists at the organizational level despite numerical gains. They suggest a need for a more comprehensive approach to empower women, which includes not only numerical representation, but also active participation of women in the organization’s decision-making process to ensure that their perspectives are heard and valued. This is illustrated in the following quotation:

“*They always thought they know about gender […] sometimes, they just think about number of participants. Like just counting the number of men and women join the session or meeting. But during the session, during a meeting, women don’t have voice to say. Women are there just like to give service […] they don’t have the voice. They just serve us water, set up things, serving people, welcoming people… For what? You can just count the number of participants. Even in a big group of women, 50 women there, and three men there, three men who have voice. For example, village chief is talking, or village health volunteers are talking, all of them are men, so only men who talk, talk, talk, while women just sit there, just for number of participants. And I think this this concept that is not understood by everyone.*”Participant 14.

*[…] “and “ມືອ່ອນເຫຍັ້ນເຫຼົ້າ—(soft hands (women) pouring liquor)”, I really want this to disappear […] still for Lao people, this is just normal. They said it’s common, it’s normal. If women are not willing to serve, or doesn’t accept to serve liquor, so they judge that you are not a good woman […] If you do something different (like not accepting to serve), you look different*”. 

Table 2 shows the presence of a distinct emphasis placed by representatives of internal organizations on the concept of “social capital”. 

#### 3.1.3. Environmental Dimension

The environmental dimension of the concept is related to the changes at the environmental level. Table 2 shows that of the six items, three items are found in all stakeholders’ concepts (“accessibility of legal services”; “quality of legal services”; and “safety of movement: perceived safety of movement outside the house”). 

The internal and external actors, as well as mothers, agree that access to basic services, such as healthcare and education, is fundamental to empowering women. In addition, they acknowledge that the quality of these services impacts women’s empowerment. Among others, healthcare centers were highlighted as the primary source of information. Therefore, the quality of services and the capacity of personnel to effectively communicate information have a significant impact on mothers’ understanding of, for instance, child nutrition, family planning, and other aspects related to maternal and child health. In addition to access to healthcare services, the internal and external actors particularly highlight that girls’ school enrollment is a key determinant of empowerment. Schools should not only be accessible for girls, but also ensure quality education. For instance, in the case of nutrition, both actors agree that integrating nutrition education and health education into the school’s curriculum is crucial to ensure that students (girls and boys), teachers, and other actors understand nutrition. Furthermore, internal and external actors emphasize the need to create a supportive environment in schools that promotes students’ health and well-being. This includes features such as school gardens, canteens with nutritious food options, school clinics, clean water, proper toilets, and handwashing facilities. According to these two actors, this supportive environment would improve healthy practices among children and ultimately lead to empowerment in the long term.

Table 2 shows that three items are found in internal and external organizations’ concepts (“social norms and stereotypes of women’s economic role in the communities in which they live (perceived by both men and women)”; “advocate change for women”; and “the ability to influence policy at the political level”). However, there is a noticeable difference in the understanding of women’s ability to influence policy decisions between the two groups. The following quotes illustrate the contrasting perceptions held by internal and external actors regarding their influence on decision making at higher positions. Internal stakeholders believe that they lack the authority to engage in plan discussions and simply adhere to their superiors’ directives. On the other hand, external stakeholders criticize the hierarchical structure, stating that they have to comply, despite disagreeing with the decisions:

Internal actor:

“*I follow the plan that they have. I am not in a position to discuss […] Because we already have a plan, we only follow. Those with higher level who take care of the project they know… we just follow the plan*”.Participant 9.

External actor:

“*The voice that can speak, it’s very hierarchy, should be really follow those steps. And also based on the position as well, right? Sometimes the staffs disagree for something, but they don’t have the right to speak out and express their opinion on that. Because they have to listen to the boss*”.Participant 14.

## 4. Discussion

The findings of the study indicate that there is a common understanding among representatives of government institutions and external organizations regarding the importance of promoting women’s empowerment to effectively reduce child malnutrition in Lao PDR. Moreover, these respondents recognize that the concept of women’s empowerment encompasses individual, relational, and environmental dimensions. The importance of addressing these various dimensions of women’s empowerment in nutrition programs was emphasized by all. At the individual level of the empowerment concept, they agree on the importance of improving women’s knowledge, leadership skills, and autonomy to act independently and of addressing gender-based violence. At the relational level of the concept, they emphasize the relevance of promoting group participation, improving household decision making, women’s control over household assets, and reducing women’s workload as a means of empowering women. At the environmental level, they express the need to address structural barriers that limit women’s access to resources and opportunities, such as access to quality basic services like healthcare and education, social norms, and stereotypes of women’s economic role in the communities, advocate change for women, and the ability to influence policy. 

However, there are some specificities regarding the understanding of the empowerment concept by both actors. Discrepancies center around two interrelated items of the conceptual framework’s concept of empowerment: women’s autonomy to act independently and social capital. The findings show that the representatives of government institutions tend to focus on the idea that women should be accompanied by the authorities to be able to act “independently”. The government has the responsibility to define and provide the education content deemed necessary to empower women. 

This likely reflects the empowerment concept of the single-party political system that constitutes the sociopolitical life of Lao PDR. The internal actors’ concept emphasizes the role of the state in addressing social and gender inequality and empowering the community (including women) [22]. 

On the other hand, the representatives of external organizations see the concept of autonomy as encompassing the capacity of women to search by themselves for the information needed to make decisions, to assert their own choices and opinions, and to decline unwanted demands. This includes the ability to reject certain social norms and practices, for instance refusing early marriage or serving alcohol to men. This perspective prioritizes human rights related to freedom to choose and decide at both the individual and community levels. It also contributed to the concern regarding the capacity of marginalized groups to express their needs [24,25,33,34]

In Lao PDR, the implementation of the social programs is influenced by a interplay between two major doctrines: the political ideology promoted by the Lao People’s Revolutionary Party (LPRP) and the “Lao Culture and Tradition (Lao way)” that give the country its specificity [35,36]. This highlights the unique nature of the Lao government’s approach to empowering women. 

Lao PDR is a single-party country, led by the LPRP. The LPRP is a communist party that promotes a Marxist–Leninist conception of society, adapted to the specificity of the cultural background of the country [23,37,38,39]. Indeed, the LPRP’s ideology is a blend of Marxist–Leninist principles and local Lao traditions and practices, which emphasizes the importance of both economic construction and Lao-specific cultural development [39,40,41,42,43,44]. In order to reach this objective, the LPRP formally gives importance to a strong, centralized government that controls the means of production and the distribution of goods and services for the benefit of its population [23]. The main means put in place by the LPRP to ensure the development of the country in line with its ideology is to promote the involvement of the party’s mass organizations in activities related to public policies, which promote the welfare of the population. Effectively, as in other communist countries, mass organizations play a significant role in the Lao political system. Mass organizations are institutions deployed in the country in order to promote the ideology among the population. In the field of nutrition, the main mass organization involved in government-approved activities is the LWU. This involvement is in keeping with the LWU’s mandate, which states that the LWU is responsible for mobilizing and uniting Lao women to protect their rights and interests, promoting gender equality and the advancement of women, educating women on government policies and laws, contributing to the protection and support of cultural traditions, and disseminating information through various media [35]. Mass organizations are supposed to be able to promote the Marxist–Leninist ideology, which is formally the basis of the LPRP. The political ideology aims to inculcate in the population values of patriotism, adherence to the one-party system, and revolutionary moral values (sacrifice, courage, intellectual rigor, solidarity with the masses, and selflessness [45]). The mass organizations must also instill in the activities they carry out the idea that the country has a cultural specificity that must be maintained. This specificity is transmitted essentially through the Three Good Deeds of the Lao Women’s Union “ຂໍ້ແຂ່ງຂັນແມ່ຍິງ 3 ດີ” that aim to empower women to contribute to the “upholding of Lao tradition and culture, and building a harmonized family environment” [36]. 

In effect, the promotion of Lao’s traditions and cultures is clearly formalized in the various policies and regulations that govern the activities of programs that include a women’s empowerment component, such as the functions and mandates of the Lao Women’s Union on the promotion and protection of Lao women’s rights [35], the Gender Equality Law, Act 11 [46], or the Prevention and Suppression of Violence Against Women and Children Law, Act 20 [47]. As a mass organization playing a significant role in women’s and children’s protection, the LWU must take into consideration the customs and traditions of the various ethnic groups [35,36,46,47]. 

From a practical point of view, the mass organizations must find a balance between egalitarian objectives promoted by Marxism–Leninism and traditions which, in Lao PDR, are strongly inspired by Confucianism, and thus by hierarchy [48]. Indeed, in Lao PDR, respect for social hierarchy and reluctance to criticize individuals with higher social positions, especially those in positions of authority, are deeply embedded in Lao tradition and culture [49]. This was clearly evident in the interviews and focus groups. When participating in nutrition programs, women tend to follow the directions given by authorities without offering criticism. For instance, the results highlight that female beneficiaries view the village LWU as role models to emulate. To promote women’s empowerment in line with the traditional and cultural value of independence, the party’s organizations are expected to serve as role models and to inspire other women to follow their lead. They can then assume the responsibility to accompany women to help them act according to the party’s conception of independence.

The findings reflect the preoccupation by the Lao government to build social capital in the country, whose foundations rest as much on the vision of the world conveyed by the propaganda of the party as they do on the traditional cultural values that characterize the various communities of the country. This approach recognizes the crucial role that both the party’s ideology and culture play in shaping Lao PDR’s social and political structure. The contribution to social capital of nutrition programs is conceived, from the government’s perspective, as being based on the networking of women who share a worldview, one promoted by the party’s ideology, while respecting ancestral traditions. The Lao government considers maintaining an equilibrium between the LPRP’s ideology and traditional cultural values crucial to preserving the country’s identity and independence in a sparsely populated country surrounded by powerful neighbors, some of whom share the communist ideology and some whom share the traditional foundations of Buddhism and Confucianism. On the other hand, external actors in nutrition programs promote a concept of empowerment that highlights the capacity of women to act autonomously. This includes the capacity to not blindly accept unwanted traditions. External actors perceive that changing certain traditional and cultural norms that have historically oppressed women represents a step towards enhancing women’s autonomy. The importance given by external actors to women’s capacity to act independently is in opposition to the message vehiculated by the Three Good Deeds dimensions promoted by the LWU. It therefore represents a challenge when having to develop and implement programs in the country. 

Although some may consider the promotion of Lao tradition and culture to be important, one has to acknowledge that it may also reinforce gender norms and stereotypes that restrict women’s opportunities and agency. For example, the emphasis on building good and prosperous families may prioritize women’s involvement in domestic and caregiving roles, which can limit their participation in the workforce and other areas of society. Additionally, traditional cultural values that prioritize hierarchy and obedience to authority over individual comfort and autonomy may reinforce gender norms and expectations that restrict women’s behavior and choices. Denes (1998) criticized these contradictions, specifically pointing out that the Three Good Deeds dimensions apply only to women and are inconsistent in their attempt to both emancipate women and reinforce traditional gender roles [50]. 

As a consequence, egalitarian objectives that characterize the political ideology are aimed at through non-radical and progressive actions that allow Confucian thought to endure in support of the health and social programs that the government is deploying in the country. When it comes to promoting social equality, it is worth noting that Confucianism could also offer some useful perspectives. Confucianism places great value on social hierarchy and respect for authority [48]. This value can play an important role in encouraging greater adherence to health and social programs’ recommendations, including those aimed at improving women’s health and well-being. For instance, the findings reveal that women tend to participate in VSLA interventions when led by the LWU because the LWU is representative of the government, hence the political authority. As such, this organization can be trusted, and its recommendations can be considered. This follow-up by women in the community is a culturally adapted strategy in the Lao context. It is therefore expected to be an effective way to empower women and to improve gender equality. When women participate in a VSLA program, as recommended by the LWU, they become more financially empowered and are better able to make informed decisions about money matters. This increased financial independence often leads to greater respect from their husbands, who recognize their ability to manage money responsibly [1], which is similar to the findings revealed from the FGD related to the GER/WWR interventions. In these interventions, the LWU encourage women to communicate in a gentle manner and express their needs in a respectful and considerate way with their husbands in order to gain support, drawing on the influence of Confucianism. Mothers who participated in GER/WWR interventions expressed that they followed the LWU’s guidance, and it had positive impacts on their families. As a result, they were able to distribute household responsibilities and have more time to care for themselves and their young children. In addition, mothers who participated in the VSLA and the GER/WWR interventions also expressed appreciation for the leadership of the LWU, which helped them to improve gender relations in the household. The findings emphasize the unique approaches of Lao PDR in promoting gender equality and women’s empowerment by integrating insights from various cultural and philosophical traditions, including Confucianism. The Lao government considers these approaches to be culturally accepted.

However, while Confucianism’s emphasis on hierarchy and respect for authority could help to encourage greater adherence to interventions aimed at promoting gender equality and women’s empowerment, it is important to note that these values might also perpetuate social and gender inequalities. The strict hierarchical structure of Confucianism might reinforce traditional respect and discourage questioning of social norms and practices [49]. Therefore, it might be worthwhile to reconsider the ways in which these traditions may perpetuate gender inequality. The existing literature shows that, despite progress in gender relations towards greater equality, achieving gender equality remains an unfulfilled goal. Women continue to be significantly underrepresented in leadership positions, and both men and women hold the perception that women lack the necessary skills for leadership roles. This perception impedes women from envisioning themselves as leaders and pursuing such roles [51,52,53]. Interviews with internal actors have further highlighted this aspect. Many female participants express that, after marriage, they hesitate to pursue their dreams due to a sense of responsibility towards their families. They perceive themselves as the ones responsible for maintaining family harmony and believe that prioritizing their careers would disrupt this balance. This perception among women, including those working with the government, may be influenced not only by the Confucian tradition but also by their aspirations to fulfill the Three Good Deeds, which emphasize the creation of a harmonized and prosperous family.

The role of tradition raises the question of how these aspects might affect the effectiveness of nutrition interventions in Lao PDR. The findings highlight that while the mothers who participated in the FGD expressed appreciation for the interventions by the LWU, one may question the truthfulness of their statements. This concern arises from the contradiction between the information provided by the same group of mothers during the FGDs and in individual interviews. The individual interviews with focus group participants present a more nuanced perspective on the necessity and perceived benefits of the GER/WWR interventions. For instance, in the individual interviews, mothers state that prior to the implementation of these interventions, they did not experience significant gender inequality issues within their families. Consequently, in their opinion, the GER/WWR interventions did not bring about substantial changes in their lives. The question regarding this contradiction may reflect the influence of tradition. According to the perspective of Confucianism, women tend to refrain from criticizing in public, particularly in front of authorities. These findings highlight the importance of government-led social interventions in empowering women, while also acknowledging that they are not the sole drivers of their empowerment. The social changes occurring in Lao PDR are likely to have a substantial impact on women’s empowerment. Rapid advancements in education, technology, and global connectivity have granted women empowerment besides what has been achieved through the government’s initiatives. Women are increasingly well informed, knowledgeable, and interconnected. They are more and more able to acquire knowledge and have the capacity to interpret the information obtained that surpasses what has been gained thanks to government programs. This could explain why some women in the community question the value of the nutrition interventions offered with the participation of the government.

Nevertheless, the concept of women’s empowerment in Lao PDR is shaped by an interplay of political and socio-cultural factors. With the government’s leading role in implementing social programs, the concept of women’s empowerment within the context of nutrition programs therefore tend to keep in line with the internal actors’ perspective that promote a “Lao way” conception of empowerment. For the time being, the external perspective on women’s empowerment may not be the best fit for nutrition programs to reduce malnutrition in the country, as there might be a not very high social acceptance of the full perspective in a country where equity between gender can still be improved. However, the coexistence of diverse approaches aimed at a shared objective incite those whose vision of the activities deployed diverge to appraise their preconceived ideas. This effect can in itself be a determinant of empowerment as it promotes a more comprehensive understanding of the complexity of the topic of interest [54,55,56]. 

Finally, we must mention that the unprecedented socio-economic changes affecting Laos are likely to have a significant impact on women’s empowerment. This underscores the need for further research to explore the impact of these changes on the concept of women’s empowerment. This would undoubtedly help us to better interpret the complex interplay between globalization and local cultural approaches in promoting improvements in gender equity and thus help us to contribute more effectively to reducing child malnutrition in Lao PDR.

## 5. Conclusions

This study aimed to provide some insights into the perspectives of different stakeholders regarding the empowerment of women in nutrition programs in Lao PDR. It shows that the Lao government integrates in its concept of empowerment respect for distinct cultural traditions that form the socio-cultural heritage of the country and give it its specificity.

## Figures and Tables

**Figure 1 ijerph-20-06662-f001:**
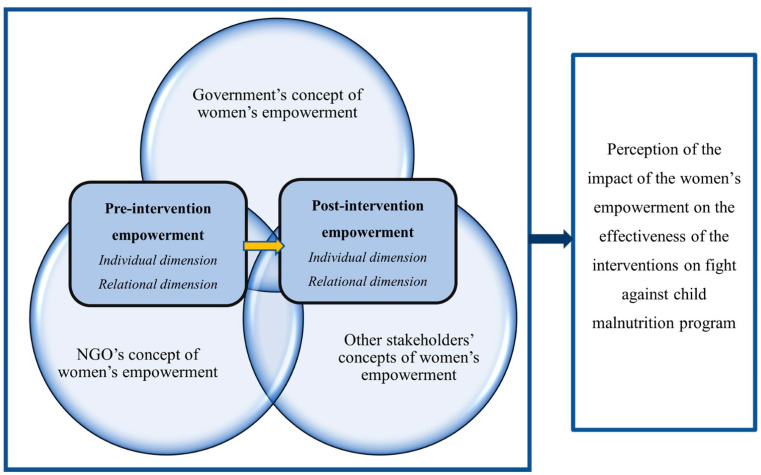
The construction of women’s empowerment regarding how to feed young children in Laos and its perceived contribution to the effectiveness of nutrition programs.

**Table 1 ijerph-20-06662-t001:** Characteristics of participants and interviews.

Populations	Total = 32
Internal organizations (government organizations)	10
External organizations	7
UN agency	1
Civil society	1
EU delegation	1
NGOs	4
Mothers of under-five-year-old children	15
**Data collection**	
In-depth interviews	Total = 20
Face to face	9
Zoom	6
Via WhatsApp	2
By phone	3
Two FGDs	Total = 15
FGD in Sensouk Village (face to face)	7
FGD in Vangneun village (face to face)	8
**Languages**	
Lao	27
English	5
**Documents**	Total = 5
Government	4
External organizations	1

**Table 2 ijerph-20-06662-t002:** The characteristics of women’s empowerment concepts held by internal organizations, external organizations involved on nutrition in Lao PDR, and by mothers of under-five-year-old children who benefit from nutrition programs in Luangprabang province. Note: + denotes the presence of the characteristic in the concept.

Individual Dimension	Int.	Ext.	Mths.
Power from within			
Self-esteem, self-confidence, and self-efficacy	+	+	+
Individual knowledge	+	+	+
Opinions (attitude and beliefs) on gender rights	+	+	+
Non-acceptability of gender-based violence (GBV)	+	+	
Opinions (attitude and beliefs) on power within the household	+	+	+
Opinions (attitude and beliefs) on freedom of movement	+	+	+
Recognition of care	+	+	+
Power over			
Individual capability	+	+	+
Personal autonomy regarding the decision to act independently		+	
Personal autonomy around violence against women (VAW)	+	+	
Access to savings	+	+	+
Access to credit	+	+	+
**Relational dimension**	**Gov.**	**Ext.**	**Mths.**
Power with			
Social capital	+		
Group participation	+	+	+
Level of support provided by groups to pursue own initiatives	+	+	+
Attitudes and beliefs of people close to the woman	+	+	
Degrees of influencing in community groups	+	+	
Participation in public events			+
Contribution to community social needs	+	+	
Power over			
Involvement in household decision making (expenditure, investment, and household management decisions)	+	+	+
Control over household assets	+	+	+
Contribution to household income: independent income	+	+	+
Power in economic pursuits (power in markets, local business)			+
Experience of GBV	+	+	
Control over time and workload	+	+	+
Ability to reduce time devoted to care responsibilities	+	+	+
Ability to redistribute the burden of care responsibilities	+	+	+
Ability to have more time for leisure and to socialize	+	+	+
**Environmental dimension**	**Gov.**	**Ext.**	**Mths.**
Accessibility of legal services	+	+	+
Safety of movement: perceived safety of movement outside the house	+	+	+
Social norms and stereotypes of women’s economic role in the communities in which they live (perceived by both men and women)	+	+	
The ability to influence at the political level	+	+	
Advocate change for women	+	+	
Quality of legal services	+	+	+

## Data Availability

The data presented in this study are not publicly available due to privacy restrictions of participants.

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
