# Peer review of "The Concepts of Women’s Empowerment in Child Malnutrition Programs in Luangprabang Province, Lao People’s Democratic Republic"

_ijerph, 2023, doi:10.3390/ijerph20176662_

Round 1
Reviewer 1 Report
Thanks for giving me opportunity for review this paper. I have several concerns regarding this paper which are given bellow:
1. The introduction needs to improve bit more briefly and define problem statement clearly in this section.
2. You mentioned “This framework has been used in many studies done in Southeast Asia in the field of nutrition and food security” in this paper, so what is new thing in your study?
3. Data collection parts is too much elaborated, please mentioned the instruments of your study in this sections as well as the interview time duration should mention here?
4. Did you follow the translation and back translation as per Braslin ?
5. Add more results table with deep rigor and make a discussion part with previous and current strong related references.
6. For your more guidance, please follow bellow research paper “ Shafiq, A., Hussain, A., Asif, M., Hwang, J., Jameel, A., & Kanwel, S. (2019). The effect of “women’s empowerment” on child nutritional status in Pakistan. International Journal of Environmental Research and Public Health, 16(22), 4499.”
7. The paper is interesting but need to do major revisions.
Moderate editing of English language required
Author Response
Dear reviewer,
We would like to sincerely thank the reviewer for the time spent reading and making highly valuable comments to the first version of our manuscript.
To facilitate the reviewers in tracking the changes in the revised manuscript, we have highlighted newly added and revised paragraphs in yellow, while deletions have been marked in red with comments alongside.
Please, find below (or the attachment) our answers to the points raised.
1. The introduction needs to improve bit more briefly and define problem statement clearly in this section.
We agree with the reviewer. The introduction could be more concise and more focused. We therefore have revised the text to this end (lines 36 – 83 in the revised manuscript).
2. You mentioned “This framework has been used in many studies done in Southeast Asia in the field of nutrition and food security” in this paper, so what is new thing in your study?
To our knowledge, no study has explored how the concept of empowerment materializes in public health programs that imply the collaboration of governmental and external organizations, that do not share the same definition of empowerment. This has been clarified (hopefully) in the introduction (line 70 to line 79 in the revised manuscript)
3. Data collection parts is too much elaborated, please mentioned the instruments of your study in these sections as well as the interview time duration should mention here?
In response to your suggestions, we have moved the mention of the instrument in the data collection tool sub-section (line 176 to line 198). We also specified the average length of interviews and focus groups (60-90 minutes) (line 173).
Recognizing that our initial methodology may have been overly elaborate, we have revised and simplified the text in order to make this section hopefully more concise while providing the information needed to judge the methodology used (lines 147 to line 172).
4. Did you follow the translation and back translation as per Braslin?
We have taken your feedback into account and have included this information in the data collection section (line 176 to line 198).
The conceptors of the Oxfam’s Conceptual Framework on Women’s Economic Empowerment do not provide a questionnaire in order to measure qualitatively the dimensions of the framework. They even suggest that researchers who want to use their framework develop questions that are adapted to the socio-cultural context of their field of study. They state that , “[…] all questionnaires should always be tested and adapted to the relevant socio-economic context under analysis, in a way that recognizes the interconnectedness of individual questions." (Lombardini, S., Bowman,. K., et Garwood, R., 2017). This is how in the literature the conceptual framework has been used. Studies on empowerment in Myanmar (Knapman, 2020), in Cambodia (Oxfam Cambodia, 2022), in Lebanon (Lombardini, Simone. & Vigneri, Marcella, 2015) for example, use different questionnaires. However, all questions are related to the dimensions of the conceptual framework. We followed their recommendation. Both tools were constructed with the participation of a member of the Lao Tropical and Public health Institute (SK) who has extensive experience in conducting nutrition research projects in the country, so that each dimension of the framework reflects the socio-cultural particularity of the study context. This is specified in line 176 to line 198
5. Add more results table with deep rigor and make a discussion part with previous and current strong related references.
We sincerely appreciate your feedback and we have added some quotations into the result section to strengthen the rigor of our study (highlighted in Yellow, line 209 – 312; line 407 – 419; line 450-466).
We also agree that the discussion should, ideally, include references about the confrontation of empowerment concepts in public health interventions led by collaborators belonging to governmental and external actors. Unfortunately, we could find any study focused on this topic We only added few more references related to the concept of women’s empowerment in general (highlighted in yellow).
6. please follow bellow research paper “Shafiq, A., Hussain, A., Asif, M., Hwang, J., Jameel, A., & Kanwel, S. (2019). The effect of “women’s empowerment” on child nutritional status in Pakistan. International Journal of Environmental Research and Public Health, 16(22), 4499.”
We appreciate your suggestion. We have consulted this reference, and it is indeed cited in our manuscript.
Once again, we extend our gratitude for your diligent review and constructive input. We are grateful for the opportunity to benefit from your expertise.
We look forward to receiving further feedback on the revised version.
Kind regards,
Kanchana Thilakoun

Reviewer 2 Report
This is a nicely written paper, with a sound methodological foundation and a well described analysis.
I find this paper raises a number of questions that are left unanswered, probably to be followed by a further publication. It seems to me as if the Lao way of addressing women’s empowerment is quite successful in providing a cultural acceptable approach, putting together several relevant aspects of this construct. The authors, however, criticize whether this approach may be effective in improving nutrition interventions for their children. Further, the authors pose the question of whether social changes occurring in Lao PDR following their one-party initiatives, or other advances in education, technology, and global connectivity may have a larger impact on women’s empowerment. Whatever is driving the change, it would be very interesting to find what impact it may have on the improvement of child’s nutrition programs, and how globalization vs. local cultural approaches bear their influence on improving gender equity.
I would recommend the authors to revise the sentences included in two consecutive paragraphs, involving lines 59-62 and 68-71, as there seems to be a repetition.
Author Response
Dear reviewer,
We would like to sincerely thank the reviewer for the time spent reading and making highly valuable comments to the first version of our manuscript. Your comment has shed light on several intriguing questions that remain unanswered, prompting us to recognize the potential for further research in these areas. In response to your suggestion, we have included this point as a suggestion for future research at the end of our revised manuscript (highlighted in yellow, line 677-683).
Furthermore, we agree with your observation regarding the repetition in two consecutive paragraphs (lines 59-62 and 68-71). We have addressed this issue by revising the sentences, removing duplication, and ensuring a smoother flow of content. The changes can be found in the revised version of the manuscript (line 58, highlighted in red with a comment alongside, please see the attachment).
Once again, we extend our gratitude for your diligent review and constructive input. We are grateful for the opportunity to benefit from your expertise.
We look forward to receiving further feedback on the revised version.
Kind regards,
Kanchana Thilakoun

Round 2
Reviewer 1 Report
Accepted